# A Hybrid Microenergy Storage System for Power Supply of Forest Wireless Sensor Nodes

**Huamei Wang, Wenbin Li \*, Daochun Xu \* and Jiangming Kan**

Key Laboratory of State Forestry and Grassland Administration on Forestry Equipment and Automation, School of Technology, Beijing Forestry University, Beijing 100083, China; whm9508@163.com (H.W.); kanjm@bjfu.edu.cn (J.K.)

\* Correspondence: leewb@bjfu.edu.cn (W.L.); xudaochun@bjfu.edu.cn (D.X.); Tel.: +86-10-6233-8139 (W.L.); +86-136-8146-2745 (D.X.)

**Abstract:** Wireless sensor nodes (WSNs) are widely used in the field of environmental detection; however, they face serious power supply problems caused by the complexity of the environmental layout. In this study, a new ultra-low-power hybrid energy harvesting (HEH) system for two types of microenergy collection (photovoltaic (PV) and soil-temperature-difference thermoelectric (TE)) was designed to provide stable power to WSNs. The power supply capabilities of two microenergy sources were assessed by analyzing the electrical characteristics and performing continuous energy data collection. The HEH system consisted of two separated power converters and an electronic multiplexer circuit to avoid impedance mismatch and improve efficiency. The feasibility of the self-powered HEH system was verified by consumption analysis of the HEH system, the WSNs, and the data analysis of the collected microenergy. Taking the summation of PV and TEG input power of 1.26 mW ($P_{PV}$:$P_{TEG}$ was about 3:1) as an example, the power loss of the HEH system accounted for 33.8% of the total input power. Furthermore, the ratio decreased as the value of the input power increased.

**Keywords:** hybrid energy harvesting; wireless sensor nodes; ultra-low-power circuit; soil-temperature-difference thermoelectric; photovoltaic

---

## 1. Introduction

In recent years, wireless sensor node (WSN) technology has developed rapidly and is widely used in environmental monitoring [1–5]. WSNs face serious power supply problems due to the complexity and particularity of the natural environment (such as in forests) and the wide distribution and large number of WSNs [3–5]. Solving the WSN power supply problem by drawing on local resources has become a research hotspot in related fields.

Efficient energy harvesting methods and techniques are becoming mature. Current research involves a variety of environmental energy sources, such as thermal energy [6–9], photovoltaic (PV) [10–12], wind energy, vibration energy [13], and so forth. However, most of the environmental energy in forests is extremely weak, making it difficult to independently supply WSNs. For instance, high canopy closure in forests results in low photovoltaic radiation, and photovoltaic energy can hardly be collected at night and on rainy days. Also, thermal energy conversion is inefficient and difficult to collect.

The collection and utilization of thermal energy technology has developed rapidly and has been studied and applied in the field of WSNs [6–9]. Mateu and Moll [6] designed an energy harvesting circuit to power sensors by using the electrical energy generated by the temperature difference between the human body and the surrounding environment. The experiments verified the feasibility of

collecting microthermal energy. Based on the intrinsic factors of soil thermal properties, a large amount of heat is stored in the soil. Huang [9] experimentally verified that the temperature difference between air and soil can be collected by a thermoelectric generator (TEG) to generate electricity for WSNs.

Photovoltaic energy is one of the most widely used types of environmental energy. Snyman and Enslin [10] designed a photovoltaic system that collects photovoltaic energy to power WSNs. However, photovoltaic collection technologies have certain limitations in the forest environment and cannot provide stable power supply for WSNs.

In addition, hybrid energy harvesting (HEH) systems have received considerable attention due to their high conversion efficiency. Using a combination of photovoltaic and thermal energy to power WSNs is efficient [14,15]. Kwan and Wu studied the dynamics and operation of a hybrid PV/TEG system and optimized the TEG design by the multiobjective NSGA-II genetic algorithm. The simulation result achieved the ideal output efficiency, which led to some electrical losses in the actual hardware design [14].

In summary, HEH systems are currently used in large-scale energy collection; however, related research about microenergy collection applied to forests is rare. Through the analysis of the characteristics of forest microenergy, we developed a low-power HEH system with automatic selection of on/off circuit functions and analyzed the power loss of each circuit part. The feasibility of this HEH system to collect microenergy and stabilize WSNs was verified.

## 2. Energy Profiles and Electrical Characteristic Analysis

### 2.1. TEG Module Characteristics

The principle of thermoelectric generation is based on the Seebeck effect, which uses thermoelectric materials to convert temperature gradients (temperature differences) into electrical energy [7]. In this study, TEGs were used to generate electricity by collecting the temperature difference between the soil temperature and the ambient temperature using a gravity heat pipe.

The core component of the TEG was a thermocouple module that connected the p-type and the n-type semiconductor material to form a pn junction [7,9]. Under the action of thermal excitation, holes and electrons in the pn junction diffused toward the low temperature port to form an electromotive force, which was the thermoelectric potential [9]. Figure 1 shows the equivalent circuit of the TEG.

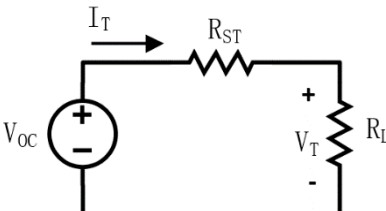

**Figure 1.** The equivalent circuit of the TEG.

The characteristic equation for this TEG module was given by [7,8], and the meaning of the formula elements is shown in Appendix A:

$$I_{TEG} = \frac{V_{OC}}{R_{ST} + R_L} = \frac{S\Delta T}{2n\frac{\rho h}{A_l} + R_L}. \tag{1}$$

The expression of the output power $P_T$ of the load $R_L$ is as follows:

As can be seen from Equation (1), the output power $P_T$ depends on the load impedance $R_L$ and the thermal resistance $R_{ST}$. When the impedance is matched (the load impedance is equal to the internal resistance), the maximum output power can be obtained:

$$P_T = I_{TEG}{}^2 R_L = S^2 \Delta T^2 \frac{R_L}{\left(2n\frac{\rho h}{A_l} + R_L\right)^2}. \tag{2}$$

$$P_{T,MPPT} = \frac{S^2 \Delta T^2}{4R_{ST}} = \frac{S^2 \Delta T^2 A_l}{8n\rho h}. \tag{3}$$

In this experimental design, the test system for soil-temperature-difference thermoelectric energy was mainly composed of a thermostatic type of gravity heat pipe (2000 × 40 mm) and eight TEGs (model TG12-8) with a physical dimension of 45 × 40 mm.

Mass-produced commercial TEGs are available, so the TG12-8 produced by Marlow Industries was selected for experimental determination. The thermostatic type of gravity heat pipe was used for thermal energy storage. The hot port of the pipe was buried at a depth of 2.5 m, and the cold port exposed to the air. The initial working state of the working medium in the pipe was liquid, which was located at the bottom (hot port) of the pipe. The phase change occurred after the working medium absorbed heat, and the liquid was converted into gas. It rose to the top of the pipe (cold port), which was converted into liquid by heat, and then folded in the bottom of the pipe to form a loop [9]. The harvester consisted of eight TEGs attached in series to the upper part of the gravity heat pipe and formed a current loop. The layout of TEGs and data transceiver modules connection is plotted in Figure 2a. The data test and transceiver modules include a DC current sensor, two data acquisition modules, and two wireless sensor transceiver modules.

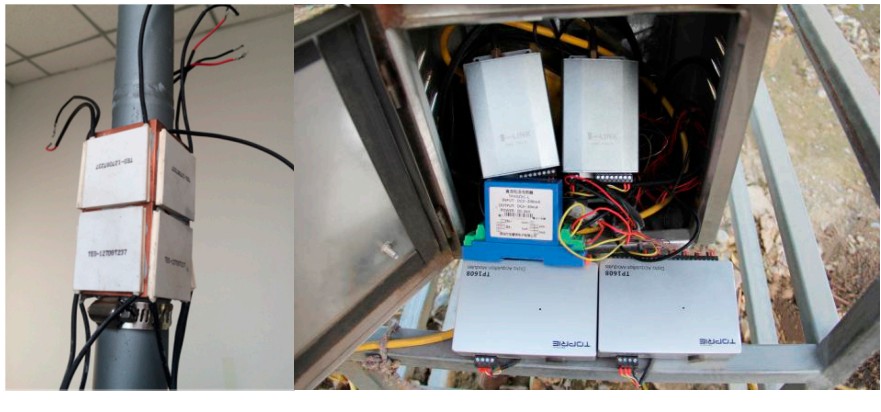

(**a**)

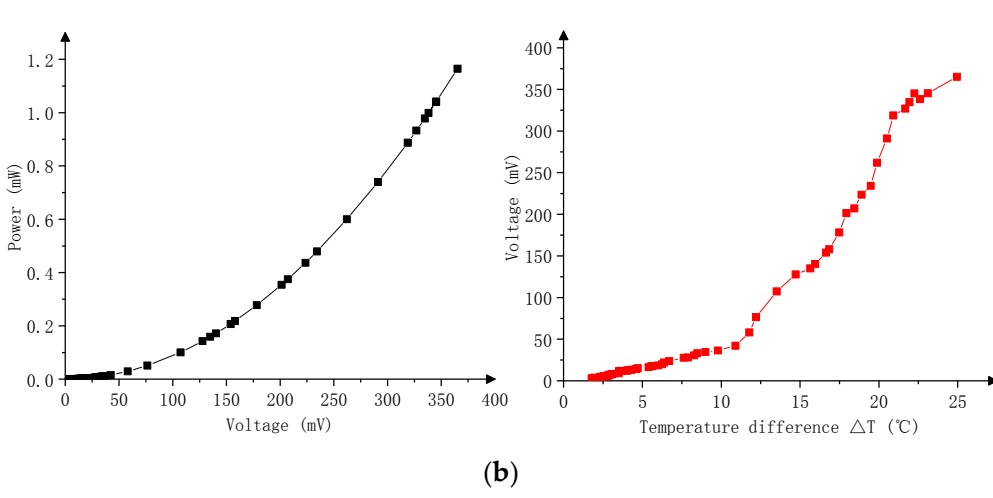

(**b**)

**Figure 2.** (**a**) Diagram of the experimental device. (**b**) Electrical characteristic plot at various temperature differences of the TEGs and data transceiver modules.

The current transmitter collected current data and the voltage measured by the data acquisition card to obtain the output power of the power-generating device (shown in Figure 2b). The sensors were connected to the data acquisition card and send the data to the Internet of Things (IoT) platform via general packet radio service (GPRS). The experimental data analysis of the IoT platform obtained the electrical characterization curves, as shown in Figure 2b.

Referring to Figure 2b, the experiment showed that the output voltage increased with the addition of temperature differences. When the value range of the temperature difference was 0–25 °C, the output voltage was approximately 0–365 mV and the output electrical power was about 0–1.17 mW.

## 2.2. PV Module Characteristics

The principle of solar power generation is the photovoltaic effect of the semiconductor interface, which converts light energy into electrical energy [10]. In the experiment, the single-crystal silicon photovoltaic panel was selected, which is characterized by high photoelectric conversion efficiency (about 15–24%) and durability. Figure 3 shows the equivalent circuit of the PV module.

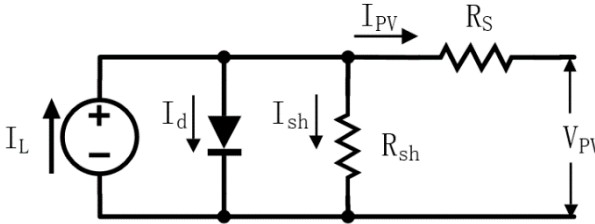

**Figure 3.** The equivalent circuit of the PV module.

The characteristic equation for this PV module was given by [11,12]:

$$I_{PV} = I_L - I_O \left[ \exp\left( \frac{V_{PV} + I_{PV} R_s}{\frac{N_s k T_t}{q} a} \right) - 1 \right] - \frac{V_{PV} + I_{PV} R_s}{R_{sh}}.$$ (4)

The calculation formula of $I_O$ and the maximum output power are as follows:

$$I_O = I_{O,n} \left( \frac{T_n}{T} \right)^3 exp\left[ \frac{q E_g}{ak} \left( \frac{1}{T_n} - \frac{1}{T} \right) \right]$$ (5)

$$PPV_{LO} \exp \frac{V_{PV} + I_{PV} R_s}{\frac{N_s k T}{q} a} \frac{V_{PV} + I_{PV} R_s}{R_{sh}} \bigg|_{PV,max}.$$ (6)

The photovoltaic test experiment used a custom 0.5 W monocrystal silicon commercial PV module (70 × 55 mm) with transparent epoxy coating and hardboard backing. The experimental measurement result of the light intensity under the condition of high canopy density was in the range of 200–1800 lux. The experimental data analysis of the electrical characterization of the PV module is plotted in Figure 4.

The experiment showed that the output voltage of the PV panel increased with load resistance. As seen in Figure 4, when the PV panel output voltage was about 3.9 V, each curve approached the maximum power points under various lighting intensities of 200, 600, 1000, 1200, 1400, and 1800 lux. The PV panel collected maximum power under different lighting conditions when the load resistance value was 8–28 kΩ. As indicated by the graph, the maximum value range of the peak power harvested by the PV panel was 0.105–2.145 mW when the PV panel output voltage was 3.9 V under photovoltaic light intensity of 200–1800 lux.

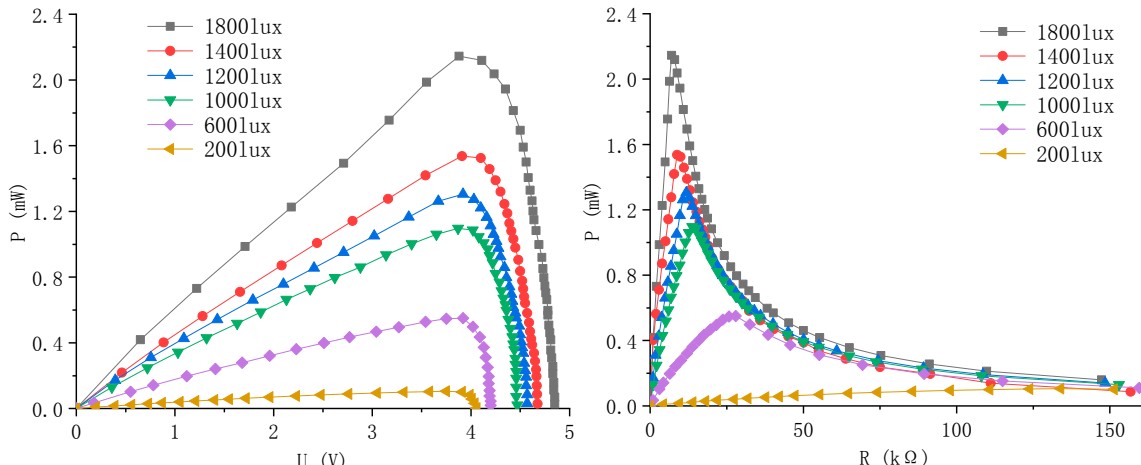

**Figure 4.** P-V and P-R characteristic plots at various lux values.

### 2.3. Power Analysis of WSNs

To study the feasibility of the hybrid energy harvesting system in this design, the low-power wireless sensor node TI eZ430-RF2500T (35 × 20 × 3.5 mm) target board was used. It consisted of the highly integrated and ultra-low-power MSP430F2274 microcontroller and the CC2500 2.4 GHz radio frequency (RF) wireless transceiver. Its operating supply voltage ranged from 1.9 to 3.6 V. The eZ430-RF2500 development tool with a wireless temperature sensor was used, and its data were transmitted every 1 s. During the TI eZ430-RF2500T execution, MSP430 and CC2500 spent most of their time in low power mode 3 (LPM3), waking up once a second to sample their ambient temperature and battery voltage; the results were sent to the network's access point (AP).

The WSNs at each execution of one operating cycle can be divided into the following eight working stages: awakened mode, temperature sample mode, message-handling mode, phase lock loop (PLL) calibration mode, receive (Rx) mode, switch between Rx and transmit (Tx) mode, Tx mode, and low-power mode. The current consumed at each stage is tabulated in Table 1.

**Table 1.** Current consumed at one working cycle of the WSN.

| Stage | Mode | Time Executed (μs) | Current Consumption (μA × s) |
|:---:|:---:|:---:|:---:|
| 1 | Awakened | 450 | 2.287 |
| 2 | Temperature sample | 115 | 3.194 |
| 3 | Message-handling | 317 | 1.382 |
| 4 | PLL calibration | 809 | 8.251 |
| 5 | Rx | 180 | 3.870 |
| 6 | Switch between Rx and Tx | 30 | 0.303 |
| 7 | Tx | 800 | 19.12 |
| 8 | Low power | 70 | 0.294 |
| | Total | 2771 | 38.701 |

The lower the frequency of data transmission, the more significant the sleep current's contribution to the average current. In a transmission period ($T_{T\_1s}$), the average current ($I_{ave}$) calculation must take into account the sleep current of the WSNs, which can be expressed as follows:

$$I_{\text{sleep}\_1s} = \left(I_{idle\_MSP430} + I_{idle\_CC2500}\right) \times \left(T_{T\_1s} - T_{exe}\right) \tag{7}$$

$$I_{\text{ave}\_1s} = \left(I_T + I_{sleep\_1s}\right)/T_{T\_1s} \tag{8}$$

where $T_{exe}$ = 2771 μs, $I_T$ = 38.701 μA×s, $T_{T\_1s}$ = 1 s, and the sum of the values (by data sheet) $I_{idle\_MSP430}$ and $I_{idle\_CC2500}$ is 1.3 μA. The calculated value of $I_{\text{sleep}\_1s}$ = 1.296 μA×s and $I_{\text{ave}\_1s}$ = 39.997 μA.

Time of execution accounted for 0.28% of the transmission period, but the current consumption of execution dominated 96.8% of the whole transmission period. So, to reduce the consumption of WSNs, it was necessary to reduce the frequency of the RF wireless transceiver operation. The RF transmission period ($T_{T\_1s}$ = 1 s) was slowed down to 60 s, which can be expressed as follows:

$$I_{\text{sleep}\_60s} = \left(I_{idle\_MSP430} + I_{idle\_CC2500}\right) \times \left(T_{T\_60s} - T_{exe}\right) \tag{9}$$

$$I_{\text{ave}\_60s} = \left(I_T + I_{sleep\_60s}\right)/T_{T\_60s}. \tag{10}$$

The calculated value of $I_{\text{sleep}\_60s}$ was 77.996 μA×s and $I_{\text{ave}\_60s}$ was 1.945 μA. The increased sleep time of an RF transmission period led to a surge in sleep current consumption. As can be seen from the above formula, the modified RF transceiver average current was 20 times lower than before. When the RF transceiver operating voltage was set to 3.3 V, the average power consumption in 1 min was approximated to be 6.419 μW. The average power consumption of the WSNs in 1 min at different working phases is shown in Figure 5.

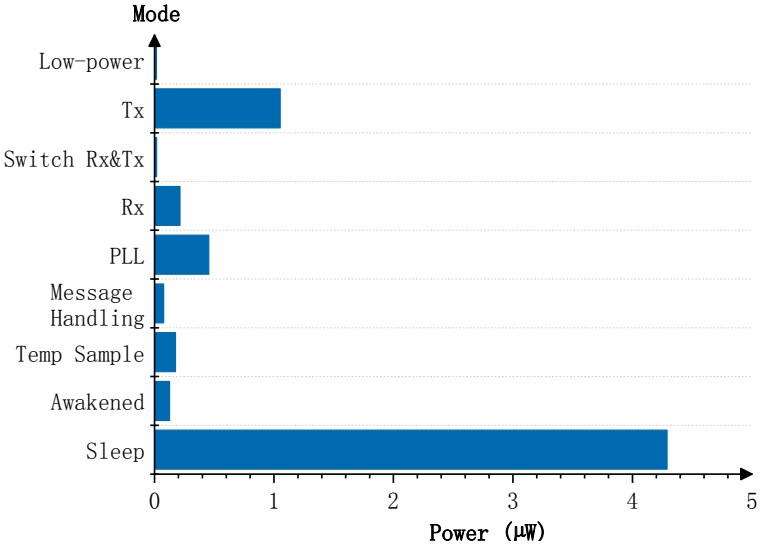

**Figure 5.** The average power consumption in 1 min at different working stages.

This altered parameter is suitable for quantities that do not rise instantly when measuring in the forest environment. During the burst transfer period, the transceiver primarily draws energy from the power management circuit to manage sudden current surges. Afterwards, the transceiver restores to sleep mode and the power management circuit is charged by the PV panel or the TEG.

## 3. Proposed System

### 3.1. Model Analysis of the HEH System

An HEH system was developed here for the purpose of avoiding inherent intermittent problems in the collection of single environmental energy sources and improving the reliability of the power supply. The core technology used is the collection of multiple energy strategies, including automatic detection and switching circuit function to improve efficiency.

HEH systems are mainly divided into three categories [16–20]:

1. Different mechanisms are designed based on the characteristics of the energy being collected. The electromagnetic and piezoelectric energy of the same vibration source is collected using a mechanically combined EH device structure, as described in [17]. There is a severe impedance mismatch between the different mechanisms, requiring different power converters to handle them separately.

2. Each EH uses a separate power converter [18]. Due to the large differences between the internal and external impedance, it is necessary to separately process the collected power using different power converters. This allows energy to be collected at the same time, but the mechanism is complex and energy loss is significant.

3. The power supply is switched on using an electronic switch [19,20]. When there are two kinds of energy at the same time, based on the priority of the power management circuit, an electronic switch is used to switch between the two energy sources, and only the higher energy is collected. The system has no impedance matching problems, but it cannot achieve simultaneous collection.

In summary, it is crucial to design an HEH system that has impedance matching and energy consumption control.

This paper mainly deals with the collection of microenergy; hence, the control of energy consumption is indispensable. Measures employed here to decrease power consumption were to reduce energy consumption by simplifying the circuit structure and to apply maximum power point tracking (MPPT) technology to improve energy collection efficiency. When designing the circuit topology, modes 2 and 3 were combined and the structure was simplified. Two independent energy conversion systems were designed to collect energy at the same time. Switching between different energy sources using an electronic switch prevents the simultaneous processing of different levels of output power that could result in damage to the system.

### 3.2. Design of the HEH System

The design features of the system are as follows:

1. According to the collection characteristics of the two energy sources, a separate energy conversion system was designed to avoid the internal impedance mismatch problem.

2. The main circuit power supply circuit line was selected as the EH (PV) system with high power and stable output. The subcircuit power supply circuit line EH (TEG) system was mainly used to charge the lithium battery and complete the supplementary power supply to the WSNs.

3. An electronic multiplexer control was used to switch the on/off of the main/subpower supply circuit line. The selection of the power supply circuit line can be adjusted according to different HEH systems, and the number of subpower supply circuit lines can be increased or decreased as needed. The HEH system's internal control is shown in Figure 6.

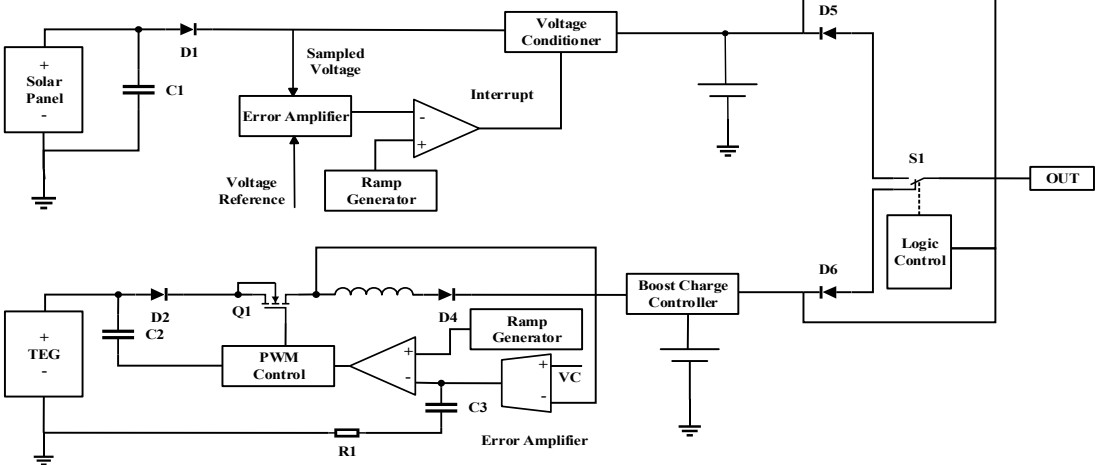

**Figure 6.** Schematic of the HEH system's internal control.

In this study, ultra-low-power chips were used to design separate energy conversion and storage for energy collection. An electronic multiplexer switching system centered on a logic switching circuit was added in order to improve collection efficiency. The design simplifies the electronic circuit structure of discrete components and replaces them with ultra-low-power integrated electronic circuits to reduce power loss.

### 3.2.1. Buck–Boost DC/DC Converter of Photovoltaic Energy

Since photovoltaic energy output is greatly affected by the environment, the energy collection of the photovoltaic panel mainly used a buck–boost DC/DC converter (LTC3119 chip) to stabilize the voltage for the core low-power voltage regulator circuit.

It can be seen from Figure 4 that each curve was near the maximum power points when the output voltage was about 3.9 V, and output power was low under the condition of high canopy density. In order to improve energy output efficiency and reduce operating consumption, the constant voltage (CV) method was used for maximum power point tracking control [11]. The output voltage of the DC/DC converter was set to 3.3 V to power the logic switching circuit. The circuit is shown in Figure 7, where P1 is the input port and P2 is the output port of photovoltaic energy.

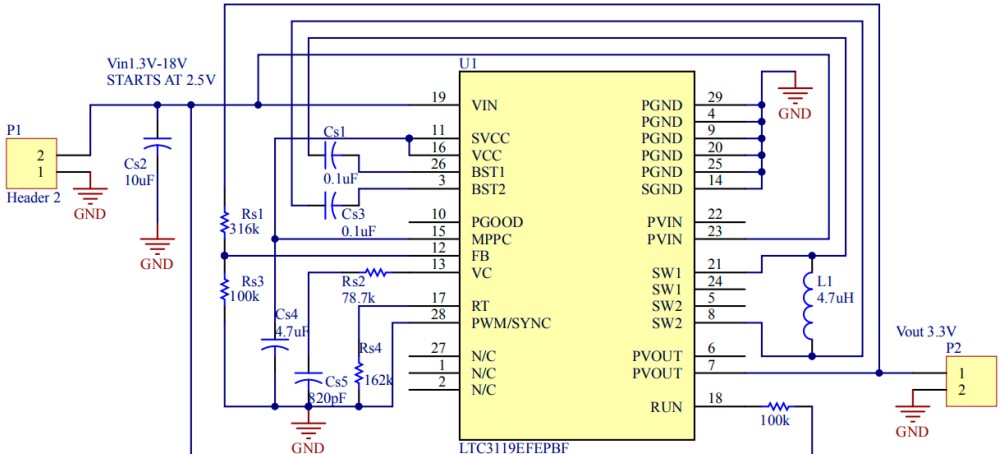

**Figure 7.** The circuit schematic of the buck–boost DC/DC converter.

### 3.2.2. Collection and Storage Circuit of Thermoelectric Energy

The soil-temperature-difference thermal energy output value was stable, but weak. An ultra-low-power boost converter and power management circuit were required to boost energy storage. The DC/DC boost converter selected in this experiment was the BQ25504 chip from TI and the power management circuit used the ultra-low-power regulator chip LTC3588-1, in combination with Li-ion/polymer shunt LTC4071 to store electrical energy in a lithium battery. The circuit is shown in Figure 8, where P3 is the input port and P4 is the output port of thermoelectric energy.

For the particularity of thermal energy, several adaptive MPPT methods suitable for thermoelectric energy harvesting have been proposed [21,22]. However, these methods have high computational power requirements and consume much energy, characteristics that are not suitable for microenergy harvesting. The resistor emulation approach was chosen for its simple structure and low power consumption. According to the literature [22], for an input power of a certain range, the DC/DC boost converter (BQ25504) with a fixed duty cycle achieves the largest output power in discontinuous conduction mode (DCM). A combined battery management system consisting of two LTC4071s can effectively improve the charging efficiency of main/sub-lithium batteries and expand storage capacity.

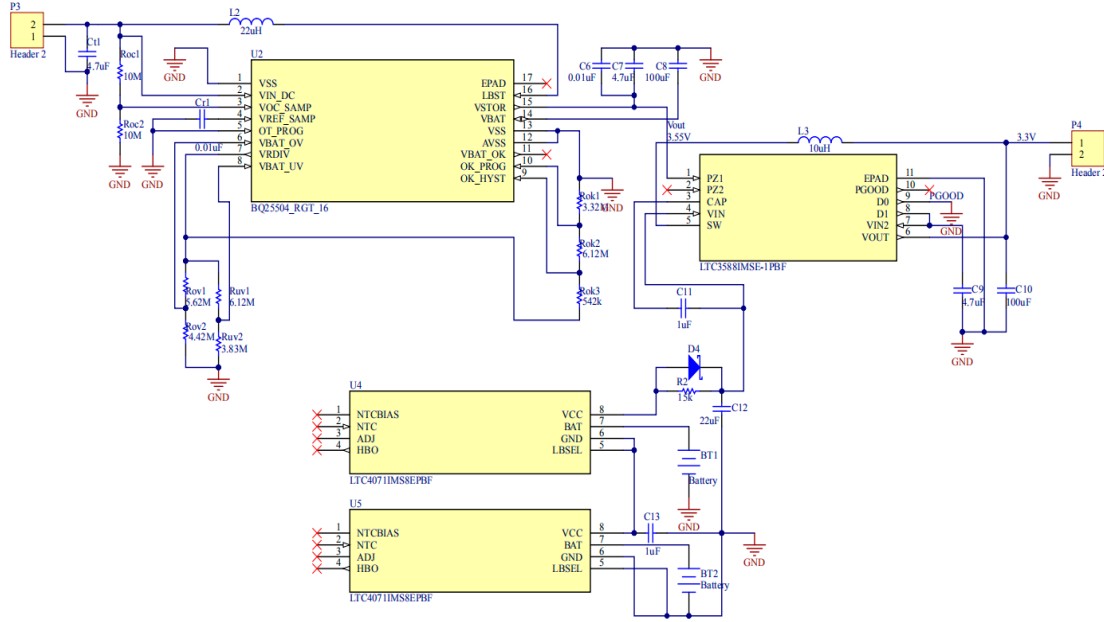

**Figure 8.** Schematic of the collection and storage circuit.

### 3.2.3. Logic Switch Automatic Conversion Circuit

In order to reduce the loss of collected energy, the electronic multiplexer TS5A3154 chip circuit was selected to switch the main/subpower supply circuit line path and the selected path supplied power for normal operation of the chip. The circuit schematic and final integrated HEH system circuit board prototype are shown in Figure 9, where P5 is the input port photovoltaic energy, P6 is the input port of thermoelectric energy, and P7 is the output port of HEH system.

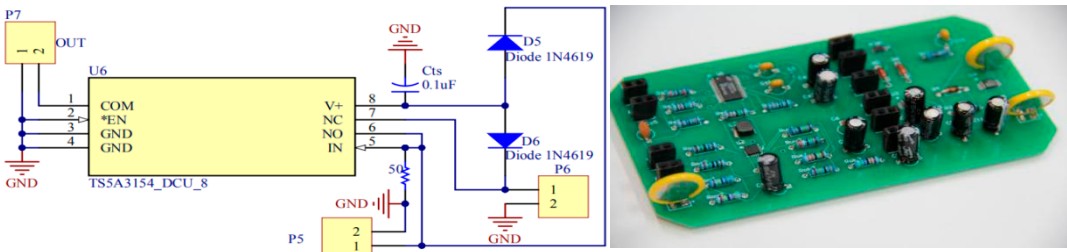

**Figure 9.** The schematic of the logic switch circuit and final integrated HEH system board.

The on/off of the logic switch was controlled by the output parameters of the two energies (output voltage and output current), for which in the initial state, the main circuit line is normally closed. The power supply of the chip was connected in parallel by two kinds of energy in order to ensure that the chip continued to work normally. When the main circuit line output voltage could not drive the chip, the logic switch switched to the subcircuit to supply power to the chip and output power. The electronic multiplexer circuit provided a stable power supply to the eZ430-RF2500T WSNs.

## 4. Experimental Results

The developed HEH system can be used to collect and store two or more types of energy (to judge the level of the electric energy order of magnitude and replace the electronic multiplexer). For the most part, the energies in forest environments are extremely faint; therefore, the loss of the circuit system has a great influence on the power collection efficiency of the system. Several experiments were conducted to analyze the feasibility of the designed HEH system and demonstrate its self-sustainable abilities.

### 4.1. Test of Thermoelectric and Photovoltaic Energy

The soil-temperature-difference electrical characteristic test experiment tests a period from September 2017 to present in Northeast China (125°42′ E, 44°04′ N). On account of the temperature change being more severe at higher-latitude areas, there is the need to more efficiently collect and utilize soil-temperature-difference thermal energy. Figure 10a shows the measured temperature difference between the environment and soil, as well as the average output power data of the IoT platform obtained for 31 January 2019 in Northeast China. The shortwave radiation data and average output power on the same day as the thermal energy test are shown in Figure 10b. In ideal conditions (without considering circuit consumption and power storage), the TEG/PV power curve obtained in the IoT platform, and HEH system average output power curve calculated from experimental data are shown in Figure 10c.

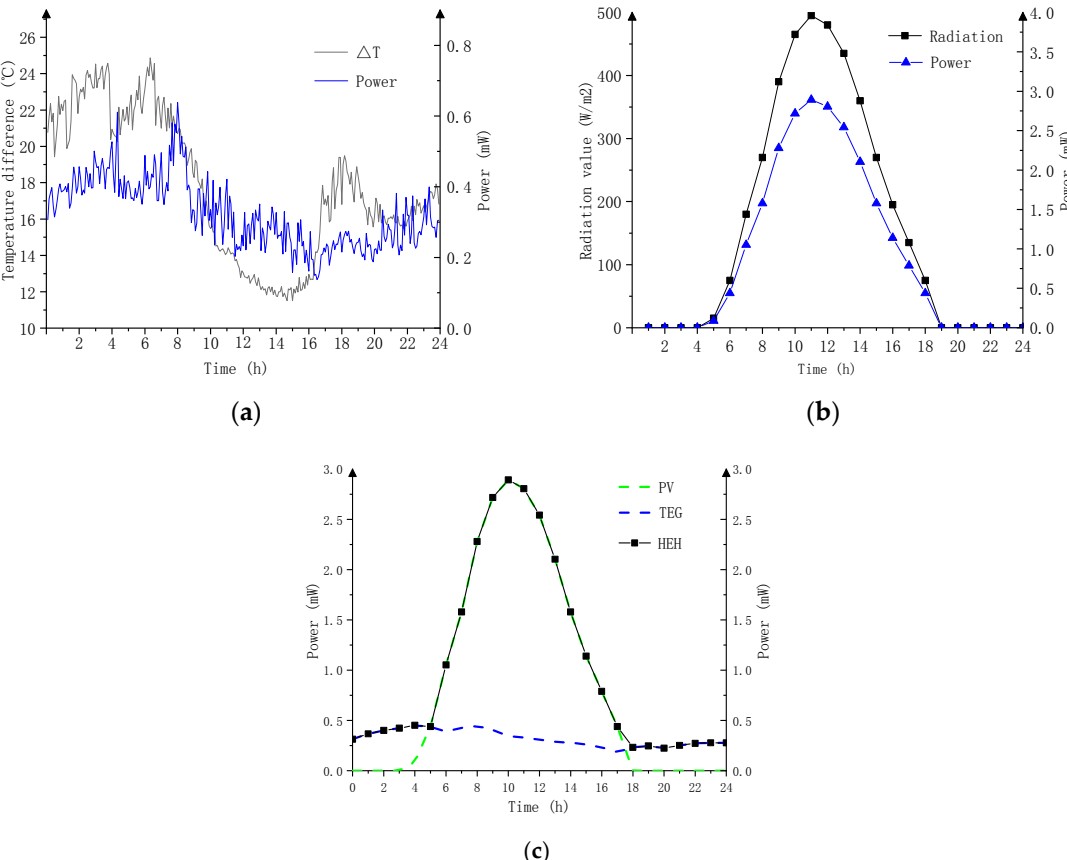

**Figure 10.** (**a**) One day's temperature difference and average output power curve, (**b**) one day's shortwave radiation and average output power curve, and (**c**) one day's average output power curve of the HEH system ideally.

The experiment site, belonging to the warm temperate zone monsoon climate, has four distinct seasons and large temperature differences. Therefore, the average output power was calculated according to the four quarters of the year; the experimental data analysis is shown in Table 2. As the ambient temperature increased, the temperature difference with the soil gradually decreased, resulting in a decrease in output power. In the first and fourth quarters, the temperature of the experimental site was the lowest in a year. The rate of soil temperature reduction was far lower than the ambient temperature, resulting in a larger soil temperature difference and electrical output power.

**Table 2.** Soil temperature difference mean and average output power for four quarters.

| Monitoring Time | Average Maximum Temperature (°C) | Average Minimum Temperature (°C) | Average Soil Temperature Difference (°C) | Average Output Power (mW) |
|---|---|---|---|---|
| First quarter (1–3) | −1.356 | −12.967 | 13.739 | 0.325 |
| Second quarter (4–6) | 20.451 | 8.989 | 7.415 | 0.221 |
| Third quarter (7–9) | 25.098 | 16.337 | 10.857 | 0.274 |
| Fourth quarter (10–12) | 2.453 | −7.424 | 13.505 | 0.372 |

In the morning and evening of the day, the temperature difference between the ambient temperature and soil temperature was large; thus, the TEG generated a greater amount of electricity. At noon, direct sunlight and fast heat transfer speeds caused the TEG to generate less electricity. However, photovoltaic systems generate peak energy at noon, not at night. Therefore, the designed photovoltaic and TEG system for complementary power generation is feasible.

The shortwave radiation data were from the Chinese National Meteorological Information Center. Data analysis showed that the solar radiation was 0 before sunrise and after sunset. Further, the average daylight duration in the first to fourth quarters were 9, 15.5, 12.14, and 8.17 h, which lasted from July 2018 to present in Northeast China. PV energy was weak in the first and fourth quarters, with the lowest average radiant energy in December, January, and February.

After powering the WSNs, the remaining PV power was stored in the battery to cope with the impact of bad weather. The battery discharged until the power was used up, and then the multiplexer switched to TEG power. The soil-temperature-difference energy was charged by the power management circuit for the main/sub-lithium battery to prevent the WSNs from unexpectedly powering off in severe weather conditions.

Combined with the electrical characteristics (Figure 10c), the following conclusion can be drawn: when the average daily (24 h) shortwave radiation is $\geq$112.3 W/m$^2$, solar energy collection can meet the normal working power supply of the WSN system during daylight hours, as calculated based on the average output power of daylight hours. Also, the thermoelectric energy would be used to supplement the power supply.

*4.2. Performance and Power Conversion Efficiency of the HEH System*

The output characteristics of the HEH circuit system are shown in Figure 11, where $V_{out}$ is the output voltage of the HEH circuit system, $I_{Out}$ is the output current of the power management circuit, and $I_{Bat1}/I_{Bat2}$ indicate the main/sub-battery output current. As shown in Figure 11a, the output voltage of the HEH circuit system was stable at 3.3 V after voltage control of the voltage regulator circuit.

The soil-temperature-difference thermal energy needs to be stored by the power management circuit to stabilize the output of electrical energy. For ideal cases, the simulated charge management circuit output current curve and the main/sub-battery output current curve are shown in Figure 11b–d. Thermoelectric energy was adjusted by the boost converter to charge the main/sub-lithium battery; therefore, $I_{Bat1}$ and $I_{Bat2}$ was increased. At the same time, the output current ($I_{Out}$) was 0.

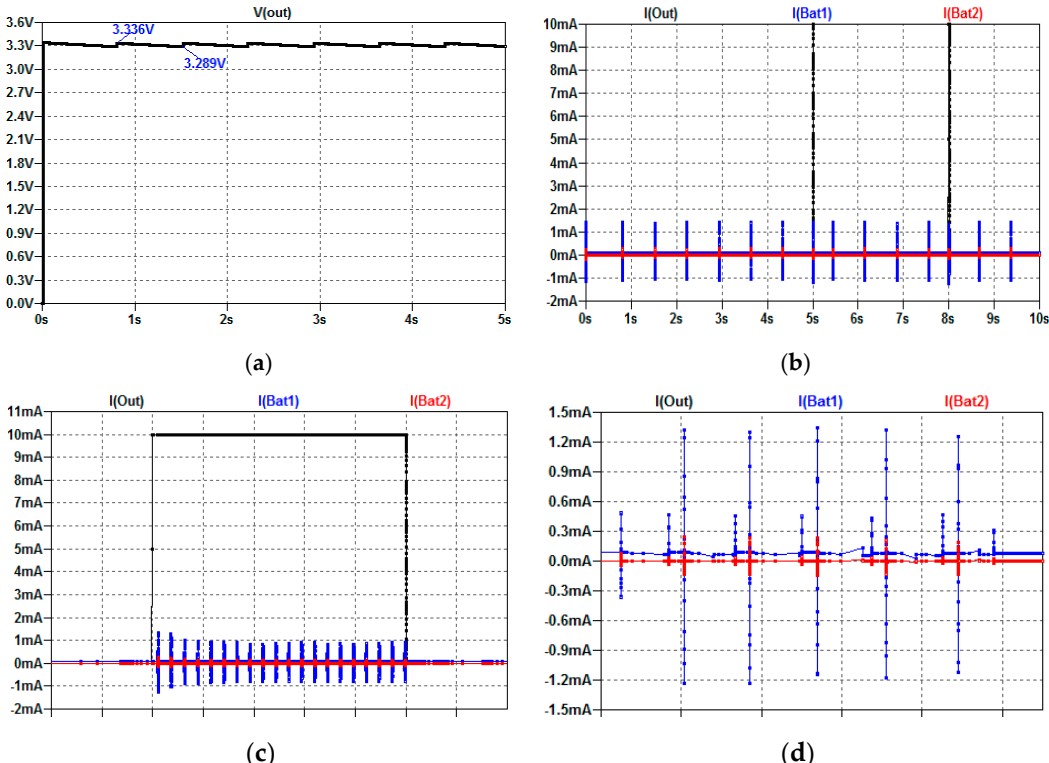

**Figure 11.** (**a**) Output voltage ($V_{out}$) curve of the HEH circuit system, (**b**) total output current curve, (**c**) output current ($I_{Out}$) curve, and (**d**) battery output current ($I_{Bat1}/I_{Bat2}$) curve.

Taking the one day's output power of PV and TEG energy as an example, the self-powering performance of HEH system was analyzed. The power consumption distribution circuit line diagram of the HEH system is shown in Figure 12.

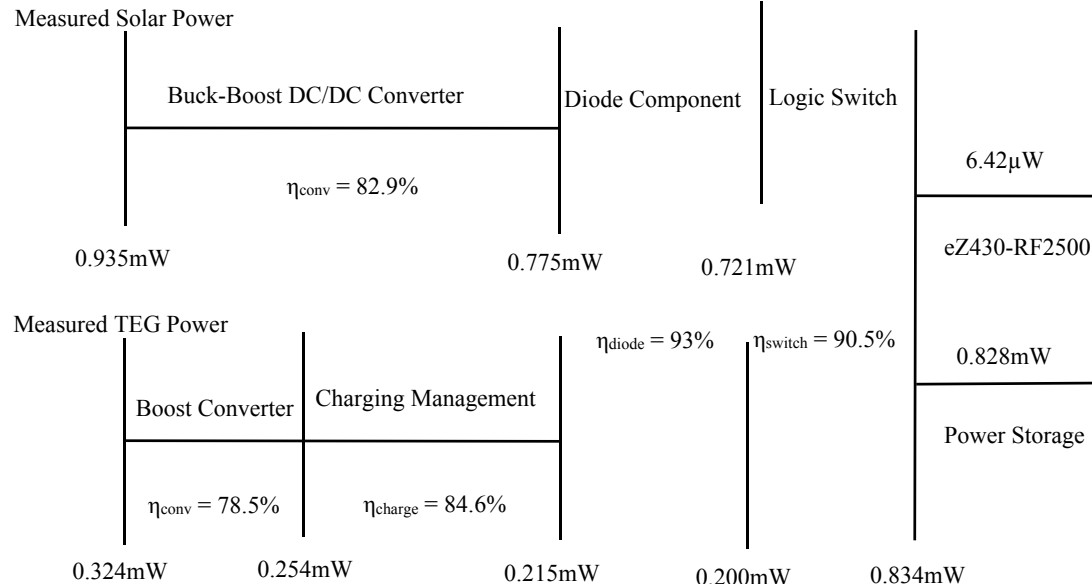

**Figure 12.** The power consumption distribution circuit line diagram of the HEH system.

The one day average output power was calculated from the test data of PV energy (0.935 mW) and TEG energy (0.324 mW), and the DC power supply (PS-305D) analog signal source was used to test the experimental circuit. As can be seen in Figure 12, 6.42 µW of electrical power was used to

power the WSNs connected to the outputs of the logic switches and 0.828 mW was left to be stored in the lithium battery separately.

It is worth noting that the PV and TEG data used in Figure 12 were the daily average output power, which was affected by the environment, and there was a case where the instantaneous power could not supply power to the WSNs, especially the PV system. Moreover, the eZ430-RF2500T used in the experiment had only one built-in temperature sensing module, and many other environmental monitoring modules are required in practical applications. Therefore, complementary circuit topology and power management circuits for both sources are required to provide stable power to WSNs.

In summary, when the average daily (24 h) shortwave radiation was >112.3 W/m$^2$, the radiant energy of the PV meets the demand for powering WSNs, and the HEH system can store most of the remaining energy. TEG power supplies the WSNs when the solar radiance is too low at night or during continuous rainy days. The HEH system can be used to compensate for a single limited form of ambient microenergy and to reduce the risk of power shortages or power outages.

## 5. Discussion and Conclusions

(1) A new low-power HEH system was designed with independent power converters that increased the stability of microenergy collection and achieved system functions of simultaneous power collection. The use of an electronic multiplexer circuit avoided impedance mismatch between different energy sources and improved efficiency.

(2) The designed HEH system implemented a self-powered system for WSNs by switching the main/subcircuit lines with logic switching devices. With flexible and self-sustainable features, it can be widely used in environmental microenergy harvesting.

(3) In the forest environment, solar energy and soil-temperature-difference thermoelectric energy have a very low level of electric energy order of magnitude. In this study, two types of environmental microenergy were collected and stored efficiently to solve the power supply problem of WSNs.

(4) The circuit topology was simplified and the MPPT method was used to reduce the HEH system's power loss. Taking the summation of PV and TEG input power of 1.26 mW ($P_{PV}$:$P_{TEG}$ was about 3:1) as an example, the power loss accounted for 33.8% of the total input power. As the value of input power increased, the ratio decreased.

The low-power processing device is selected in this design, which leads to the design limitation of the system's poor processing performance against sudden surges and strict requirements for input energy parameters. The collection strategy of the HEH system can be universal, but the collection of different energy requires customized device selection and MPPT method selection. Subsequent research should focus on overall optimization and versatility of the system and further reduction of the power loss of the electronic system to provide a self-sustaining and stable supply of power to WSNs.

**Author Contributions:** All authors contributed to the paper. W.L. provided the required components and contributed research ideas. H.W. carried out most of the research work. W.L., D.X., and J.K. reviewed and corrected the manuscript.

**Funding:** The authors gratefully acknowledge the financial support from the National Science Foundation of China (31670716), the China Postdoctoral Science Special Foundation (2016T90044), and the China Postdoctoral Science Foundation (2015M570945).

**Acknowledgments:** The authors would like to thank Beijing Forestry University, the National Science Foundation of China, the China Postdoctoral Science Special Foundation, and the China Postdoctoral Science Foundation for providing the necessary funds and the conducive environment to carry out this research.

**Conflicts of Interest:** The authors declare no conflict of interest.

## Appendix A

**Table A1.** Nomenclature, abbreviations, and numerical constants.

| Nomenclature | | | |
|---|---|---|---|
| $V_{OC}$ | open circuit voltage of TEG (V) | $I_{O,n}$ | nominal saturation current (A) |
| $I_{TEG}$ | loop current of TEG (A) | $T$ | actual temperature of PV module (°C) |
| $R_{ST}$ | thermal resistance of TEG (Ω) | $T_n$ | nominal temperature of PV module (°C) |
| $R_L$ | load resistance (Ω) | $E_g$ | the band gap energy of the semiconductor |
| $S$ | Seebeck coefficient of TEG (V/K) | $I_{sleep\_1s}$ | sleep current in 1 s of WSN (μA×s) |
| $\Delta T$ | temperature difference between the hot port and the cold port (°C) | $I_T$ | total current consumption in a transmission period of WSN (μA×s) |
| $n$ | number of thermocouples | $I_{idle\_MSP430}$ | idle current of MSP430 of WSN (μA) |
| $\rho$ | resistivity of the material (Ω×mm) | $I_{idle\_CC2500}$ | idle current of CC2500 of WSN (μA) |
| $h$ | height of a single thermocouple leg (mm) | $T_{T\_1s}$ | transmission period in 1 s of WSN (s) |
| $A_l$ | area of a single thermocouple leg (mm) | $I_{ave\_1s}$ | average current in 1 s of WSN (μA) |
| $I_d$ | reverse saturation (dark) current of the PV module diode (A) | $T_{exe}$ | total execution time for MSP430 and CC2500 of WSN (s) |
| $P_T$ | output power of TEG (W) | Abbreviations | |
| $P_{T,MPPT}$ | maximum output power of TEG (W) | TEG | thermoelectric generator |
| $I_L$ | light-generated current (A) | IoT | Internet of Things |
| $I_{PV}$ | output current of PV module (A) | GPRS | general packet radio service |
| $V_{PV}$ | output voltage of PV module (V) | PV | photovoltaic |
| $R_s$ | series resistance of PV module (Ω) | WSN | wireless sensor node |
| $R_{sh}$ | shunt resistance of PV module (Ω) | RF | radio frequency |
| $T_t$ | temperature of the pn junction (°C) | HEH | hybrid energy harvesting |
| $a$ | diode ideality constant | Numerical constants | |
| $N_S$ | number of cells | $q = 1.60217646 \times 10^{-19} C$ one electron charge | |
| $I_O$ | diode saturation current of PV module (A) | $k = 1.3806503 \times 10^{-23} J/K$ Boltzmann constant | |

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
