# Peer review of "A Hybrid Microenergy Storage System for Power Supply of Forest Wireless Sensor Nodes"

_electronics, doi:10.3390/electronics8121409_

Round 1
Reviewer 1 Report
The paper gives a good introduction into the HEH. The high value of the work is the combination of theoretical and experimental approach, however the experimental test rig is rather poorly presented. Please replace Figure 2a with a more informative one and give a more precise description of the test rig.
Please indicate how the temperature is transferred from the 2 m depth - heat pipe?
Please indicate clearly which are modeling and experimental results (e.g. figure 10).
Author Response
Dear Reviewer:
We are grateful for your valuable comments and suggestions, which helped us improve the quality of the paper. The comments are thought-provoking and helpful for our research work. We have studied the comments carefully and made modifications and corrections. Responses to the comments are as follows. (The line number described in the reply is based on the PDF file.)
Point 1: Please replace Figure 2a with a more informative one and give a more precise description of the test rig.
Response 1: We already changed Figure 2a with the photograph of data test and transceiver modules (Fig.2a in Page 3) and added the description (Line 98-100 in Page 3).
Point 2: Please indicate how the temperature is transferred from the 2 m depth - heat pipe?
Response 2: We added the description of the heat transfer process through the heat pipe. (Line 91-95 in Page 3)
Point 3: Please indicate clearly which are modeling and experimental results (e.g. figure 10).Please indicate clearly which are modeling and experimental results (e.g. figure 10).
Response 3: We added comments to all data plots.(Figure 2b_ Line 106 in Page 3; Figure 4_ Line 127 in Page 5; Figure 10_ Line 283-284 & Line 286-287 in Page 10; )
We have fully checked the manuscript again and made revisions as shown in “track changes”. Furthermore, the full manuscript was typeset again.
We are thankful for your comments on our manuscript. We believe the revised version of our manuscript provides clearer descriptions of the main points of our study.
Thank you for your consideration.
Best wishes,
Wenbin Li
School of Technology
Beijing Forestry University
Beijing 100083, China
Email: leewb@bjfu.edu.cn
Reviewer 2 Report
This is an interesting study. Research background was well described in the Introduction section with relevant references. The methodology and results are appropriate. I recommend publication after a few minor revisions.
(1) Some parts in Sub-sections 2.1 and 2.2 are so basic. There can be replaced with citations.
(2) In the Discussion and Conclusion section, there exists some summary of this study. More discussion is required with considerations of limitations and future works.
Author Response
Dear Reviewer:
We are grateful for your valuable comments and suggestions, which helped us improve the quality of the paper. The comments are thought-provoking and helpful for our research work. We have studied the comments carefully and made modifications and corrections. Responses to the comments are as follows. (The line number described in the reply is based on the PDF file.)
Point 1: Some parts in Sub-sections 2.1 and 2.2 are so basic. There can be replaced with citations.
Response 1: Some basic theoretical introductions have been deleted or replaced with citations. (Line 67-69 & Line 71-75 in Page 2 & Line114-117 in Page 4)
Point 2: In the Discussion and Conclusion section, there exists some summary of this study. More discussion is required with considerations of limitations and future works.
Response 2: We added the relevant content of limitations and future works in the Discussion and Conclusion section. (Line 372-377 in Page 14)
We have fully checked the manuscript again and made revisions as shown in “track changes”. Furthermore, the full manuscript was typeset again.
We are thankful for your comments on our manuscript. We believe the revised version of our manuscript provides clearer descriptions of the main points of our study.
Thank you for your consideration.
Best wishes,
Wenbin Li
School of Technology
Beijing Forestry University
Beijing 100083, China
Email: leewb@bjfu.edu.cn
Reviewer 3 Report
The paper describes a hybrid microenergy storage system for power supply WSN in forest environnment. Authors want to highlight that
the originality of the reseach work is based on the use of hybrid energy harvesting by using photovoltaic and thermoelectric
based on soil-temperature-difference and on the air. Authors use a lot of references in order to prove the originality of they
research works. But after reading, some remarks can be made about th whole of the article :
- the solar and thermoelectric energy are described extensively in the scientific literature, the authors bring no new theory on the subject;
- assembling electronic modules is not really a research job but more engineering;
- the energy consumption described in paragraph 2.3 is not very clear. To make a power budget, the size of the data to
be sent must be taken into account;
- the datasheet of the radio module gives a current consumption of several tens of mA. I have a doubt about the ability
of the energy harvesting system to provide enough current to the radio module.
Overall, the paper needs to be completely rewritten by focusing on the actual feasibility of the system.
Author Response
Dear Reviewer:
We are grateful for your valuable comments and suggestions, which helped us improve the quality of the paper. The comments are thought-provoking and helpful for our research work. We have studied the comments carefully and made modifications and corrections. Responses to the comments are as follows. (The line number described in the reply is based on the PDF file.)
Point 1: the solar and thermoelectric energy are described extensively in the scientific literature, the authors bring no new theory on the subject.
Response 1: The innovation of this paper focuses on the original design and feasibility test of a hybrid energy harvesting (HEH) system. In Sections 2.1 and 2.2, the electrical characteristic analysis and experimental test of PV and TEG module was to determine the input parameters of the HEH system and to complete the circuit design. Subsequent research should focus on combining the theoretical calculation with the verification of the experimental data to study the new theory of HEH system.
Revision 1: Some basic theoretical introductions have been deleted or replaced with citations. (Line 67-69 & Line 71-75 in Page 2 & Line114-117 in Page 4)
Point 2: assembling electronic modules is not really a research job but more engineering.
Response 2: In Sections 3.2.1-3.2.3, the electrical topology designs of each module circuit and the measures of implementation function were described in detail, in order to ensure the integrity of the paper and the repeatability of the experiment. The design of overall schematic (as shown in Figure 6) and the each module circuit(as shown in Figure 7-9) were introduced in Sections 3.1 and 3.2. Figure 12 was a power loss line graph for each part of the circuit used to verify the feasibility of the system.
Revision 2: We will be most grateful if you would like to make more specific revise opinion.
Point 3: the energy consumption described in paragraph 2.3 is not very clear. To make a power budget, the size of the data to be sent must be taken into account.
Response 3: In Section 2.3, we added the relevant content of the WSN transmission data size. During the test, the WSN only transmitted the data of temperature and battery voltage. We changed the data transmission frequency from once per second to once per minute, to reduce the power loss.
Revision 3: We added the relevant content of the WSN transmission data size.(Line 144-146 in Page 5)
Point 4: the datasheet of the radio module gives a current consumption of several tens of mA. I have a doubt about the ability of the energy harvesting system to provide enough current to the radio module.
Response 4: Table 1. showed the working time of the WSN at each execution and the corresponding current consumption in one operating cycle. The average loss current in 1s was 39.997µA which was calculated by Equation (7)-(8). In order to increase the insomnia time and reduce the overall power consumption, the WSN data transmission frequency was set to 1 min and the average power consumption was approximated to be 6.419 µW.
Based on the input parameters (U/I/P of Solar and TEG), the output parameters (power loss of the WSN) and the power loss analysis of the circuit system, the feasibility of the HEH system in this design was demonstrated (as shown in Figure 12).
We have fully checked the manuscript again and made revisions as shown in “track changes”. Furthermore, the full manuscript was typeset again.
We are thankful for your comments on our manuscript. We believe the revised version of our manuscript provides clearer descriptions of the main points of our study.
Thank you for your consideration.
Best wishes,
Wenbin Li
School of Technology
Beijing Forestry University
Beijing 100083, China
Email: leewb@bjfu.edu.cn
Round 2
Reviewer 3 Report
I would like thanks the authors to their answers to my questions. The newly submission describes more clearly the problematic of the HEH system particularly in the forest environment. Authors concentrated their works on the optimisation by using dual EH with ultra-low power components. Conclusions show the limitations of the system and suggest possible extensions of the system with other energy harvesting sources.
The article is not really original but worth reading.